# Parametric Surface Modelling for Tea Leaf Point Cloud Based on Non-Uniform Rational Basis Spline Technique

**DOI:** 10.3390/s21041304

**Published:** 2021-02-11

**Authors:** Wenchao Wu, Yongguang Hu, Yongzong Lu

**Affiliations:** Key Laboratory of Modern Agricultural Equipment and Technology, Ministry of Education Jiangsu Province, Jiangsu University, Zhenjiang 212013, China; wuwenchao1993@126.com (W.W.); luyongzong@126.com (Y.L.)

**Keywords:** leaf point cloud, surface fitting, principal component analysis, slice, particle swarm optimization

## Abstract

Plant leaf 3D architecture changes during growth and shows sensitive response to environmental stresses. In recent years, acquisition and segmentation methods of leaf point cloud developed rapidly, but 3D modelling leaf point clouds has not gained much attention. In this study, a parametric surface modelling method was proposed for accurately fitting tea leaf point cloud. Firstly, principal component analysis was utilized to adjust posture and position of the point cloud. Then, the point cloud was sliced into multiple sections, and some sections were selected to generate a point set to be fitted (PSF). Finally, the PSF was fitted into non-uniform rational B-spline (NURBS) surface. Two methods were developed to generate the ordered PSF and the unordered PSF, respectively. The PSF was firstly fitted as B-spline surface and then was transformed to NURBS form by minimizing fitting error, which was solved by particle swarm optimization (PSO). The fitting error was specified as weighted sum of the root-mean-square error (RMSE) and the maximum value (MV) of Euclidean distances between fitted surface and a subset of the point cloud. The results showed that the proposed modelling method could be used even if the point cloud is largely simplified (RMSE < 1 mm, MV < 2 mm, without performing PSO). Future studies will model wider range of leaves as well as incomplete point cloud.

## 1. Introduction

Plant leaves show different 3D architecture during growth [1], and their spatial appearance also shows sensitive responses to environmental conditions, such as drought stress [2,3,4], cold stress [5], and light availability [6]. Additionally, the morphological features of leaf influences nutrients resorption [7] and canopy capacity, such as rainfall interception [8,9] and light interception [10]. Moreover, leaf morphological features can be utilized in the analysis of leaf shape diversity of a species [11] and crop breeding [12]. In addition, a previous study [13] has found a quadratic function relationship between lettuce fresh weight and leaf surface areas estimated from convex and concave hulls, which could be used for estimation of crop yield with further investigation. Some morphological characteristics, such as leaf length, leaf width, leaf area, specific leaf area, leaf inclination angle, and leaf bend angle, have been directly taken as research objects among these studies. However, almost no consideration was given to three dimensional morphological traits, which can be used to accurately describe leaf shape and can provide more nuanced understandings to leaf’s dynamic adaptions to environmental conditions.

To describe leaf’s spatial shape and its changes, it is necessary to obtain 3D architecture information of leaf surface. Nowadays, devices like depth camera, stereo camera and laser scanner can be easily used to obtain point cloud of plant and its organs. Over the last few years, leaf and stem classification of plant point cloud [14,15,16,17,18] has received an immense amount of attention, whereas few studies in the literature have been published specifically on parametric modeling for plants and their organs. Modeling for leaf point cloud is a challenging problem due to wide variations in leaf morphology caused by species diversity.

General meshing or remeshing methods, such as advancing-front mesh [19], Delaunay triangular mesh [20], and meshes combined with simplification or refinement algorithms [21,22] could be used to fit leaf point cloud. Oqielat [23,24] proposed a kind of surface fitting method for plant leaf by combining the Clough-Tocher method with radial basis functions. Although fitted meshes could represent leaf shape well, it is difficult to describe complicated spatial features of leaf surface with mathematic expressions. Parametric surface modelling technologies could fit a leaf point cloud into a spatial surface, by which leaf’s geometric features can be easily described. For example, leaf edge can be represented as spatial curves, whose geometric parameters such as perimeter, the first derivative, the second derivative, and curvature, and so on, could be calculated to describe leaf’s morphological changes. Wen et al. [25] proposed a leaf modelling method based on hierarchical representation of veins and margin, which were constructed with B-spline curves, but this method is only applicable for leaf point clouds with obvious venation structure. Earlier studies [26,27] have realized visualization and reconstruction of plant leaves using Bezier and B-spline surface; however, no metrics were specified to evaluate fitting error, and important parameters like control points or interpolated points were selected manually. Additionally, Beardsley and Chaurasia [28] reconstructed plant leaf by using Bezier curves to represent salient leaf components (midrib, silhouettes, and cross-section), but this method considered these components separately and thus resulted in loss of surface’s information.

In summary, accurate parametric leaf surface model could be used to describe leaf morphology and further describe leaf’s adaptation process to environment; however, few relevant studies have been conducted until now. Non-uniform rational B-spline (NURBS) is applicable to construct free-form curves and surfaces of real-world objects, and it was adopted in this study. Additionally, tea leaf was taken as the modeling object; the obtained results can be applied to similar shaped leaves. Figure 1 illustrated the overview of our proposed method for parametric leaf modeling.

The aim of this study is to provide a parametric surface modeling method for accurately fitting tea leaf point cloud into NURBS surface. There are two key problems in the method. One is to generate the point set to be fitted (PSF) from leaf point cloud, the other is to fit the PSF into precise NURBS surface. Two methods were proposed to generate different PSFs based on principal component analysis (PCA) and point cloud slicing technique. Additionally, the PSF was firstly fitted by B-spline surface and then was transformed to NURBS form by adjusting weights of control points by minimizing fitting error, which was defined as a multi-objective optimization problem, and particle swarm optimization (PSO) was employed to solve the problem.

## 2. Materials and Methods

### 2.1. Data Acquisition

Experimental tea leaves were obtained from two species of *camellia sinensis* (*zhongcha* 108 and *Maolv*) in Maichun tea plantation at Danyang, Jiangsu province, China (32°02′35″ N, 119°67′80″ E), in October 2020. As shown in Figure 2, morphologically from top to bottom, there are apical bud, real leaves, and fish leaf in sequence on a tea stalk. Real leaves were collected as experimental object of this study. In the experiment, a 12 m×5 m rectangle sampling zone was chosen for each tea species, and each zone lay across 8 lines of tea trees as each line is 1.5 m wide. Each sampling zone was divided into 40 sampling plots, namely the involved tea trees in each line were divided into 5 plots, and one leaf was sampled from each plot. 40 real leaves of each tea species, were picked down, removed petioles, and scanned to get point clouds by EXAscan 3D scanner (CREAFORM INC., Lévis, QC, Canada). For avoiding leaf shape changes largely, the leaves were picked and scanned one by one, and the entire process consumed less than 5 min for each leaf.

### 2.2. Data Pre-Processing

The scanned point cloud data inevitably included noise points, therefore, denoising and extracting the portion belonging to leaf is the precondition before modeling. Since pre-processing of the point cloud is not the key problem of this study, MeshLab software (version 2020.12) was used to remove redundant points of backgrounds to obtain pure leaf point cloud.

### 2.3. Parametric Surface Modelling

The parametric surface modeling process contains two tasks: generating the PSF and NURBS surface modeling. In the section that follows we briefly present how to generate the PSF from leaf point cloud and how to fit the PSF into NURBS surface. To begin the process, the PCA was used to adjust the position and posture of the point cloud. Then, the point cloud was sliced into multiple sections, and several slices were selected to generate the PSF. Finally, the obtained PSF was fitted into NURBS surface.

#### 2.3.1. Principal Component Analysis of Leaf Point Cloud

To avoid unnecessary complexity caused by difference in position and posture of leaf point clouds in the process of generating PSF, PCA is used to find three principal components of individual leaf point cloud, and then the point cloud is transformed into a standard position and posture, which is called as standard leaf point cloud (SLPC). As shown in Figure 3, the center of the SLPC is located at the origin of the coordinate system, and the first, second and third principal components correspond to *x*-direction, *y*-direction, and *z*-direction of the coordinate system, respectively. All following treatments are implemented on the SLPC.

#### 2.3.2. Point Cloud Slicing and Slice Selection

Generally, it is unnecessary to construct a surface interpolating thousands of points due to considerations of the computation cost and the precision limitation of measuring devices. Therefore, slice technique is adopted to divide the SLPC into multi-sections, and part of slices are selected to generate the PSF.

To be specific, the SLPC is sliced along the direction of the first principal component, as shown in Figure 4a. Supposing that number of slices $num_{s}$ is determined, then slice’s thickness $W_{s}$ can be calculated as:(1)$$W_{s}=\frac{L}{num_{s}}$$
where L is leaf length. Centre plane, perpendicular to the first principal component and defined as projection plane, of a slice was calculated as:(2)$$PL_{i}=min\left(x\right)+\left(i-0.5\right)\times W_{s},i=1,\cdots ,\;num_{s}$$
where *x* is the coordinates of the SLPC in the first principal component.

To detect leaf base and leaf tip, two additional projection planes $PL_{0}$ and $PL_{end}\left(PL_{num_{s}+1}\right)$ together with corresponding slices $Slice_{0}$ and $Slice_{end}\left(Slice_{num_{s}+1}\right)$ are created on corresponding positions.
(3)$$PL_{0}=min\left(x\right),\;PL_{end}=max\left(x\right)$$
Thicknesses of $Slice_{0}$ and $Slice_{end}$ are set to zero, and they are overlapping with $PL_{0}$ and $PL_{end}$, respectively. As shown in Figure 4b, $Slice_{0}$ comprises three feature points $P_{l0}$, $P_{m0}$ and $P_{r0}$ belonging to left edge, midrib, and right edge, respectively. While $Slice_{num_{s}+1}$ also comprises three feature points, and they are overlapping and denoted as $P_{end}$.

To determine three feature points in $Slice_{0}$, supposing there is a slice ranging from leaf base, increasing its thickness till the range of the y-coordinates of the slice’s points is not less than 2 mm, which is average diameter of tea leaf petiole. The minimum value and the maximum value of the y-coordinates of the slice’s points are set as the *y*-coordinates of $P_{l0}$ and $P_{r0}$, respectively. The mean value of *z*-coordinates of the slice’s points is set as *z*-coordinates of $P_{l0}$ and $P_{r0}$, and the *x*-coordinates of $P_{l0}$ and $P_{r0}$ are set as min(x). $P_{m0}$ is calculated as midpoint between $P_{l0}$ and $P_{r0}$. Besides, $P_{end}$ is the point of the SLPC in the $Slice_{end}$, and mean value would be calculated if there are more than one point in the slice.

To avoid excessive computation, part of slices is selected to generate the PSF. Note that the selected slices should distribute uniformly in the SLPC to contain as much surface information as possible, and the slices $Slice_{0}$ and $Slice_{end}$ must be included.

#### 2.3.3. Generating the Point Set to Be Fitted

Margin and midrib are distinct geometric features of tea leaf, and they can be extracted as framework of leaf surface. However, the framework is insufficient to represent spatial architecture of tea leaf, so points belonging to leaf surface should be taken as consideration together. Consequently, the PSF consists of points belonging to leaf framework and leaf surface in each selected slice. Two methods are developed to generate an ordered PSF, and an unordered PSF from l+1 selected slices, respectively. The ordered PSF consists of a point set $\left\{Q_{i,j}\;|\;j=0,\cdots ,\;l;\;i=0,\cdots k;\;l,k>2\;\mathrm{and}\;l,k\in N^{*}\right\}$ by extracting k+1 points from each of the l+1 slices, while the unordered PSF consists of a point set $\left\{Q_{i,j}\;|\;j=0,\cdots ,\;l;\;i=0,\cdots k_{j};\;l,k_{j}>2\;\mathrm{and}\;l,k_{j}\in N^{*}\right\}$, in which the number, $k_{j}$, of points extracted from each slice could be different.

For generating an ordered PSF, $k=3+t_{l}+t_{r}\;\left(t_{l},\;t_{r}\in N^{*}\right)$ points are selected from each selected $Slice_{i}\left(i\neq 0,\;num_{s+1}\right)$, as shown in Figure 5a. Firstly, three feature points, $P_{li}$, $P_{mi}$ and $P_{ri}$ belonging to left edge, midrib and right edge, are found in each slice. Specifically, the three points are leftmost, highest, and rightmost points in a smaller region $T_{i}$ inside $Slice_{i}$, and the region lays on the center of the slice and its thickness is $\frac{W_{s}}{2}$. Then, $t_{l}$ equally spaced points between $P_{li}$ and $P_{mi}$, and another $t_{r}$ equally spaced points between $P_{mi}$ and $P_{ri}$ are calculated, respectively. Finally, corresponding nearest points of these points in the slice are selected.

For generating an unordered point set, a different method is developed, in which all points in a selected $Slice_{i}\left(i\neq 0,\;num_{s+1}\right)$ are projected vertically on the corresponding center plane, as shown in Figure 5b.

Additionally, same points are obtained in $Slice_{0}$ and $Slice_{num_{s}+1}$ for both the ordered PSF and the unordered PSF. Specially, for $Slice_{0}$, three feature points, $t_{l}$ equally spaced points between $P_{l0}$ and $P_{m0}$, and $t_{r}$ equally spaced points between $P_{m0}$ and $P_{r0}$ are selected directly. Additionally, there are $3+t_{l}+t_{r}$ overlapping points $P_{end}$ in $Slice_{num_{s}+1}$.

#### 2.3.4. Accuracy Evaluation for Fitted Surface Model

To better understand the procedure for fitting obtained PSF into NURBS surface and assess accuracy of fitted surface model, definition of NURBS surface and metrics of accuracy evaluation for fitted surface are introduced in this section.

A NURBS surface of degree (p,q) with (n+1)×(m+1) control points is defined as a bivariate vector-valued piecewise rational function of the form:(4)$$S\left(u,v\right)=\frac{\sum_{i=0}^{n}\sum_{j=0}^{m}N_{i,p}\left(u\right)N_{j,q}\left(v\right)\omega_{i,j}P_{i,j}}{\sum_{i=0}^{n}\sum_{j=0}^{m}N_{i,p}\left(u\right)N_{j,q}\left(v\right)\omega_{i,j}},\;a\leq u,v\leq b$$
where, u and v are two independent parameters; $P_{i,j}$ is the (i,j)th control point in three-dimensional space; $\omega_{i,j}$ is corresponding weight of $P_{i,j}$; $N_{i,p}\left(u\right)$ and $N_{j,q}\left(v\right)$ are the ith and jth non-rational B-spline basis functions of *p*-degree and *q*-degree defined on knot vectors $U=\left[u_{0},\;u_{1},\cdots ,\;u_{n+p+1}\right]$ and $V=\left[v_{0},\;v_{1},\cdots ,\;v_{m+q+1}\right]$, respectively. Specifically,
(5)$$\begin{array}{c} N_{i,0}\left(u\right)=\{\begin{array}{c} 1,u_{i}\leq u<u_{i+1}\\ 0,\;\mathrm{otherwise}\; \end{array}\\ N_{i,p}\left(u\right)=\frac{u-u_{i}}{u_{i+p}-u_{i}}N_{i,p-1}\left(u\right)+\frac{u_{i+p+1}-u}{u_{i+p+1}-u_{i+1}}N_{i+1,p-1}\left(u\right) \end{array}$$
(6)$$U=\left[\underset{p+1}{\underbrace{a,\cdots ,a}},\;u_{p+1},\cdots ,\;u_{n},\;\underset{p+1}{\underbrace{b,\cdots ,b}}\right]$$

In this study, domain of both parameters u and v are set as [0, 1]. The surface is also known as B-spline surface if all weights $\omega_{i,j}$ are set as 1.

It would be argued that geometric model of tea leaf should be built to describe its morphological features as much as possible. Therefore, Euclidean distance between fitted surface and the SLPC could be used to evaluate the accuracy of fitted surface.

To decrease computation cost, an ordered test subset was obtained to represent the SLPC by method described in Figure 5a. To calculate Euclidean distance between a point G and a surface, it is equal to find a point $G^{\prime }$ on the surface, which satisfies the distance between the two points is smaller than distance between the point G and any other points on the surface. Newton iteration method [29] was used to determine $G^{\prime }$, see Appendix A. The initial values of parameters u and v of $G^{\prime }$ are calculated as:(7)$$u_{1}=\frac{∥G-S\left(0,0.5\right)∥}{∥G-S(0,0.5)∥+∥S\left(1,0.5\right)-G∥}$$
(8)$$v_{1}=\frac{∥G-S\left(u_{1},0\right)∥}{∥G-S\left(u_{1},0\right)∥+∥S\left(u_{1},1\right)-G∥}$$
Additionally, the root-mean-square error (RMSE) is calculated as:(9)$$RMSE=\sqrt{\frac{\sum_{i=1}^{n}dist_{i}^{2}}{n}}$$
where $dist_{i}$ is distance between the ith point and fitted surface. Furthermore, a weighted sum of the RMSE and maximum value (MV) of the distances between the subset and the fitted surface is prescribed as error metric:(10)$$F=\alpha_{1}RMSE+\alpha_{2}MV$$
where $\alpha_{1}$ and $\alpha_{2}$ are subjected to $\alpha_{1}+\alpha_{2}=1$.

#### 2.3.5. Modeling the Point Set to Be Fitted

In this study, the PSF is fitted with B-spline surface firstly and then is transformed to NURBS form. B-spline surface fitting is performed to determine knot vectors and control points of the surface, then multi-objective optimization is employed to adjust weights of control points according to accuracy requirements.

For an ordered PSF $\left\{Q_{i,j}\;|\;j=0,\cdots ,\;l;\;i=0,\cdots k;\;l,k>2\;\mathrm{and}\;l,k\in N^{*}\right\}$, it can be fitted by a (p,q)th-degree B-spline surface with (m+1)×(n+1) control points (p≤m, q≤n). It satisfies m=k and n=l in the case of interpolation, while m<k or n<l in the case of approximation. Two popular algorithms, respectively for B-spline surface interpolation and approximation, can be employed to fit the ordered PSF; however, they are unavailable for an unordered PFS due to the number of points in each selected slice are not equal. Therefore, a different fitting algorithm is proposed, which can be applied to fit not only the ordered PSF but also the unordered PSF. Main steps of the new algorithm and the general algorithms are shown in Algorithm 1 and Algorithm A1 (Appendix B), respectively. The general algorithms use fixed parameterized values $\bar{u}\;\left(\bar{v}\right)$ and knot vector U (V) when performing B-spline curve fitting for each of columns (rows) of the PSF, while the new algorithm uses different parameterized values and knot vectors when performing B-spline curve fitting.
**Algorithm 1:** A New B-spline surface fitting algorithm**Input: **Q, p, q, m, n**Output: **U, V, P1: **function**
GetUVP(Q, p, q, m, n)
2: // (1) Perform B-spline curve approximation through l + 1 columns of points of Q (*u*-direction)3:        **for**
j = 0 → l
**do**4:        temp0 ← {Q[0][j], Q[1][j], …, Q[$k_{j}$][j]}5:        Parameterize temp0 to obtain ${\bar{u}}^{j}$ by Equations (A4), (A5), or (A6)6:        Calculate knot vector $U_{j}$ using ${\bar{u}}^{j}$ by Equation (A8)7:        Fit temp0 into *p −degree* curve to obtain control points $R_{j}$ using ${\bar{u}}^{j}$ and $U_{j}$ ▷ $R_{j}$ consists of  m+1 points8:        **end for**9:        R ← {$R_{0}$,$\;R_{1}$, …,$\;R_{l}$} ▷ R consists of (m+1)×(l+1) points10:        Calculate mean value of each elements of $U_{0}$, $U_{1}$, *...,*
$U_{l}$ to get knot vector U
11: // (2) Perform B-spline curve approximation through m + 1 rows of points of the R (*v*-direction)12:        **for**
i = 0 → m
**do**13:        temp1 ← {R[i][0], R[i][1], ..., R[i][l]}14:        Parameterize temp1 to obtain ${\bar{v}}^{i}$ by Equations (A4), (A5), or (A6)15:        Calculate knot vector $V_{i}$ using ${\bar{v}}^{i}$ by Equation (A8)16:        Fit temp1 into *q−degree* curve to obtain control points $P_{i}$ using ${\bar{v}}^{i}$ and $V_{i}$ ▷ *P_i_* consists of n+1 points17:        **end for**18:        P ← {$P_{0}$,$\;P_{1}$, …,$\;P_{m}$} ▷ P consists of (m + 1)×(n + 1) points19:        Calculate mean value of each elements of $V_{0}$, $V_{1}$, *...,*
$V_{m}$ to get knot vector V
20:        **return**
U, V, P
21: **end function**

To improve the fitting accuracy, it is equal to decrease the error metric F. Once control points, knot vectors, or weights of control points changes, fitted surface will change accordingly. In this study, weights of control points are adjusted to reduce fitting error, and an objective function F(X) is used:(11)$$F\left(X\right)=\alpha_{1}RMSE+\alpha_{2}MV$$
where X is the weights matrix of the surface, $\alpha_{1}$ and $\alpha_{2}$ are subjected to $\alpha_{1}+\alpha_{2}=1$.

PSO, a kind of swarm intelligence algorithms proposed by Kennedy and Eberhart [30], is used to solve F(X). In the algorithm, every potential solution is regarded as a particle with its own position X and velocity V in multiple dimensional variable space. Additionally, N particles, respectively with positions of $X_{1}^{0},\;X_{2}^{0},\cdots ,\;X_{N}^{0}$ and velocities of $V_{1}^{0},\;V_{2}^{0},\cdots ,\;V_{N}^{0}$, are randomly generated in initialization stage. Each particle adjusts its velocity and position as Equations (12) and (13) until reaching the optimal solution:(12)$$V_{i}^{j}=\theta V_{i}^{j-1}+c_{1}r_{1}\left(P_{best,i}-X_{i}^{j-1}\right)+c_{2}r_{2}\left(G_{best}-X_{i}^{j-1}\right)$$
(13)$$X_{i}^{j}=X_{i}^{j-1}+V_{i}^{j}$$
where i=1,⋯, N, $V_{i}^{j}$ and $X_{i}^{j}$ is the velocity and position of the ith particle in jth iteration, and the best position of the ith particle and best position of all particles up to current iteration are denoted as $P_{best,i}$ and $G_{best}$, respectively. $c_{1}$ and $c_{2}$ are the non-negative learning rates representing the influences of cognitive (individual) ability and social (group) ability, respectively. $r_{1}$ and $r_{2}$ are uniformly distributed random numbers in the range [0,1], which increasing the randomness of particles’ movement. θ is inertia weight and is assumed to decrease linearly as Equation (14), which can ensure both strong capabilities of global search in early stage and local search at late stage [31].
(14)$$\theta =\theta_{max}-\frac{\left(\theta_{max}-\theta_{min}\right)\times j}{j_{max}}$$
where $\theta_{max}$ and $\theta_{min}$ are 0.9 and 0.4 in this study, j and $j_{max}$ are current iteration number and maximum iteration number, respectively.

## 3. Results

The scanned point clouds consist of 33,000 to 50,000 points by over-scanning 80 experimental tea leaves. The leaf length ranged from 45 to 85 mm along the first principal component direction. According to observation of underside of experimental tea leaves, the leaf shape could be roughly divided into two categories. One looks like a hill, in which the midrib is a distinct ridge line. The other includes those leaves, which looks like a smoother convex, or is flatter, or with wrinkles on leaf surface, a common feature is that leaf midrib is difficult to extract.

In this section, a case study, illustrating the whole process of the proposed parametric surface modelling method for tea leaf point cloud, was demonstrated in detail. Furthermore, the impacts of point’s number in point cloud on the fitting accuracy were investigated.

### 3.1. Case Study of the Proposed Parametric Surface Modeling Method

The whole process to modelling leaf point cloud into a parametric surface includes 2 steps: generate the PSF and fit the PSF into NURBS surface. A case study was demonstrated as following to show how a PSF was generated and how it was fitted into a NURBS surface. 

#### 3.1.1. Generating an Ordered PSF and an Unordered PSF

Firstly, a tea leaf point cloud was transformed to a SLPC by PCA, and it could be referred to Figure 3. Then, the SLPC was sliced into 50 slices, and another two slices were created on the leaf base and leaf tip, respectively, as shown in Figure 6a. Next, 7 slices, $\left\{Slice_{i}|i=0,\;9,\;17,\;26,\;34,\;43,\;51\right\}$, evenly distributed in the SLPC were selected to generate an ordered PSF (Figure 6b) and an unordered PSF (Figure 6c). Total 7×7 points were generated for the ordered PSF, by selecting 7 points from each of selected slices as Figure 5a. While all points in a selected slice were projected onto corresponding projection planes for the unordered PSF. It should be noticed that in both PSFs, there were same points in $Slice_{0}$ and $Slice_{51}$, respectively.

#### 3.1.2. Fitting the PSF into NURBS Surface

Either the two general algorithms or the new algorithm, B-spline curve fitting was performed across each of slices of the PSF at first. Figure 7 showed different fitting results of points in the 4th selected slice of Figure 6b,c, and degrees of all curves are 2. Figure 7a showed an interpolated curve and an approximated curve through 7 points, the two curves have 7 and 5 control points, respectively. Additionally, Figure 7b showed an approximated curve, with 5 control points, through all projected points. Notice that two fitted curves were in space rather on a plane in Figure 7a, while the fitted curve in Figure 7b was on the projection plane.

Take the unordered PSF in Figure 6c as example, the new algorithm was adopted to fit the PSF into a (2,2)th B-spline surface with 5×5 control points. As illustrated in Figure 8a, B-spline curve fitting was performed in u-direction for each of slices of points of the PSF firstly. Then, the obtained point set, composed of the control points of the fitted curves in u-directions, were fitted in v-direction, and in the meantime control net of the B-spline surface were obtained, as shown in Figure 8b. Finally, the fitted B-spline surface could be obtained, as seen in Figure 8c.

Visually, the obtained B-spline surface well fitted the SLPC, as illustrated in Figure 8c. For quantitatively describing fitting error, a test subset composed of 90 points, evenly distributed on the leaf surface, were selected from the SLPC. The subset was obtained by dividing the point cloud into 10 slices with 9 points selected in each slice (selected as Figure 5a, and satisfied $3+t_{l}+t_{r}=9,\;t_{l}=t_{r}=3$). The RMSE and MV of distances between the subset and fitted surface shown in Figure 8c were 0.31 and 1.08 mm, respectively. To reduce the fitting error, a PSO multi-objective optimization process by minimizing F(X)=0.8RMSE+0.2MV, was conducted to transform the fitted B-spline surface into NURBS form. Additionally, the particle number and iterative number were set as 5 and 10, respectively. The RMSE and MV were reduced to 0.25 mm and 1.04 mm, respectively.

### 3.2. Impacts of Point’s Number in Point Cloud on Fitting Accuracy

For studying the influences of points’ number in point cloud on fitting accuracy, 8 samples were randomly chosen from 80 experimental point clouds. By using voxel grid filter [32], the number of points of the 8 sample point clouds were down-sampled to 300, 500, 1000, 2000, 4000, 8000, 16,000 and 32,000 with 10 repetitions, respectively. Every point cloud was sliced into 10 sections, and 8 slices were selected to generate unordered PSF, which was fitted into surfaces of degree (2,2) with (3 × 7) control points. The RMSE and MV of Euclidean distances between fitted surfaces and subsets (selected just like the test subset described in Section 3.1.2 from SLPC) were calculated. Figure 9 showed the fitting errors of fitted B-spline surfaces of simplified point clouds, and each RMSE or MV value was calculated by mean value of 10 repeating groups. The results showed that the RMSE values were always less than 1 mm, while the MV values were less than 2 mm, as the point’s number ranged from 300 to 32,000. It is worth nothing that the MV usually occurs on the edge near the leaf base or tip. Additionally, fitting error could be further reduced by PSO when transforming B-spline surface into NURBS form.

## 4. Discussion

The parametric surface modelling method proposed in this study can accurately reconstruct 3D architecture of leaf point cloud with NURBS surface. The method included two essential steps: generating the PSF and fitting the PSF into NURBS surface.

Based on PCA and slicing technique, two methods were developed to generate ordered PSF (i.e., (r+1)×(s+1) points) and unordered PSF (i.e., a point set composed of r+1 columns of unequal number of points), respectively. Although the ordered PSF was obtained based on feature points on left edge, midrib, and right edge of leaf, all points were selected automatically instead of manually [26], and there was no need for extracting points belonging to the secondary and tertiary veins [25]. Moreover, the unordered PSF was not affected by the feature points, and it can be generally applied to other plant leaves. In addition, both PSFs included information of leaf surface as well as leaf edge and midrib.

A new B-spline surface fitting algorithm was proposed for the unordered PSF. Compared to the general interpolation and approximation algorithms, the new algorithm can be used to fit not only the ordered PSF but also the unordered PSF, while the general algorithms only can be applied to the ordered PSF. Previous studies ended up with Bezier or B-spline surface fitting [26,33], and the fitting error was fixed. However, a multi-objective optimization process, solved by PSO, was employed in this study to reduce fitting error by adjusting weights of control points, and the B-spline surface was transformed to NURBS form in the meantime.

Compared to meshing and remeshing methods, such as Delaunay triangulation, edge collapse [34], vertex removal [35], vertex clustering [36], our method just generates PSF from point cloud without complicated operations on points themselves. Additionally, in the end, our method obtains a NURBS surface, while in essence all meshing or remeshing methods obtain a set of points, which connect to other according to special rule.

The proposed parametric surface modelling method works well even if there are few points in the point cloud, as illustrated in Figure 9. However, some limitations need to be acknowledged. First, feature points belonging to leaf midrib were not easy to be extract sometimes, which results in failure to generate the ordered PSF, and an unordered PSF must be chose in this situation. Secondly, the leaf tip was simplified as a point in this study, which caused maximum fitting error occurs on the leaf edge. Thirdly, this study made no attempt to reconstruct leaf point cloud in incomplete situation as well as the leaves’ point cloud with other shapes. Additionally, the study would have been more interesting if including intelligent methods to determine the number and selection of slices, the degree of NURBS surface, and the number of control points. More research is needed to account for these questions.

The proposed parametric surface modelling method has several practical and potential applications: (1) monitoring changes in 3D architecture of plant leaves under both suitable growth conditions and abnormal growth conditions, such as suffering from drought, salty, cold, diseases, and pests, etc., which provides a basis to quantitatively evaluate the impacts of environmental stress, diseases, or insect pests on plants; (2) studying the diversity of species or genetic expression of crops, which can be used to explore plants’ adaptability to the environment or promote the breeding process of crops; (3) simulating the leaf growth process in virtual plants by adjusting control points or weights of NURBS surface, which easily changes leaf shape; (4) combining the parametric surface model with the mechanical model to investigate dynamic leaf responses to rainfall, droplets, and wind, or interaction with agricultural and horticultural machinery; (5) analyzing the canopy capability of a plant, such as light or rainfall interception.

## 5. Conclusions

In this study, a parametric surface modelling method was proposed for tea leaf point cloud. Firstly, the principal component analysis is used to adjust leaf point cloud into a standard position and posture. Then, the obtained point cloud is sliced into multiple sections, and some slices are selected to generate the point set to be fitted. Next, the point set to be fitted is fitted as B-spline surface. Finally, the fitted B-spline surface is transformed to NURBS form by particle swarm optimization for improving fitting accuracy. The fitting error is described as the Euclidean distance between fitted surface and leaf point cloud.

The principal component analysis operation decreased difficulty of subsequent slicing operation brought by differences in shape, spatial posture, and position of leaf point cloud. According to 3D structure of tea leaf, two methods were proposed to generate two types of points to be fitted: ordered and unordered. The latter could be easily obtained than the former since the former relies on distinguished framework of leaf margin and midrib in point cloud. In addition, a new B-spline surface fitting method was developed for the latter, and it could be employed in fitting the former as well.

Preliminary study demonstrated that the proposed method could accurately reconstruct tea leaf point cloud even if the point’s number in point cloud decreased sharply. Additionally, as illustrated in Figure 9, no distinct increases occurred on the root-mean-square error and maximum value of Euclidean distances between fitted B-spline surfaces and test subsets of point clouds, as the point’s number decreased from 32,000 to 300. It could be observed that all values of RMSE and MV were smaller than 1mm and 2mm, respectively, and the values could be further decreased by particle swarm optimization. Future research should assess the influences of slice number, surface degree, and number of control points on fitting accuracy. It should be noticed that the maximum distance between fitted surface and the test subset usually appears near the leaf base or tip, which may be resulted from simplifications of them in the fitted surface. Therefore, further studies should improve accuracy of the leaf model, especially in the leaf base and tip. The proposed method could be applied in other plant leaves with similar shape to tea leaf. Additionally, the methods for PSF generation and NURBS surface fitting should be improved for satisfying needs of wider range of plant types. Moreover, modelling an incomplete leaf point cloud should be conducted in the future.

## Figures and Tables

**Figure 1 sensors-21-01304-f001:**
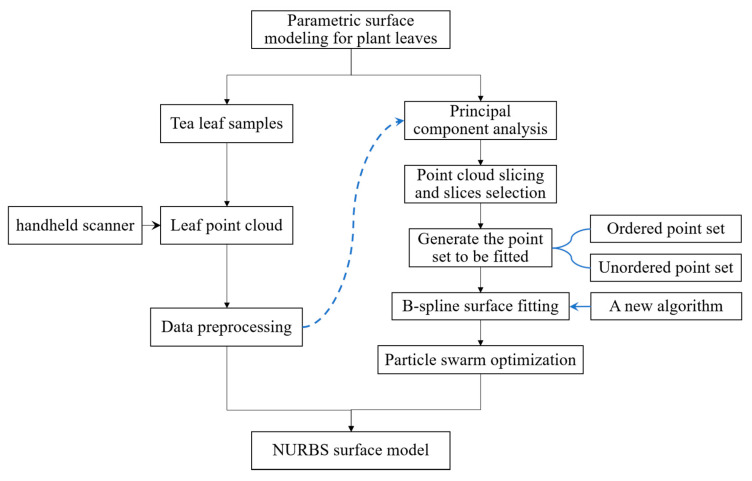
Overview of the proposed method for parametric surface modeling.

**Figure 2 sensors-21-01304-f002:**
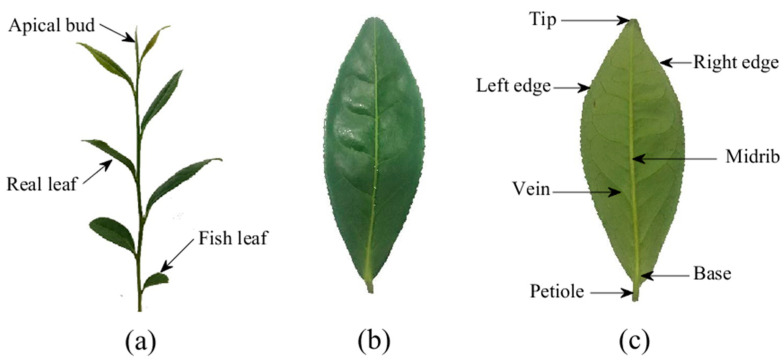
(**a**) A tea stalk with several leaves; (**b**) Upper side of a real leaf; (**c**) Underside of a real leaf.

**Figure 3 sensors-21-01304-f003:**
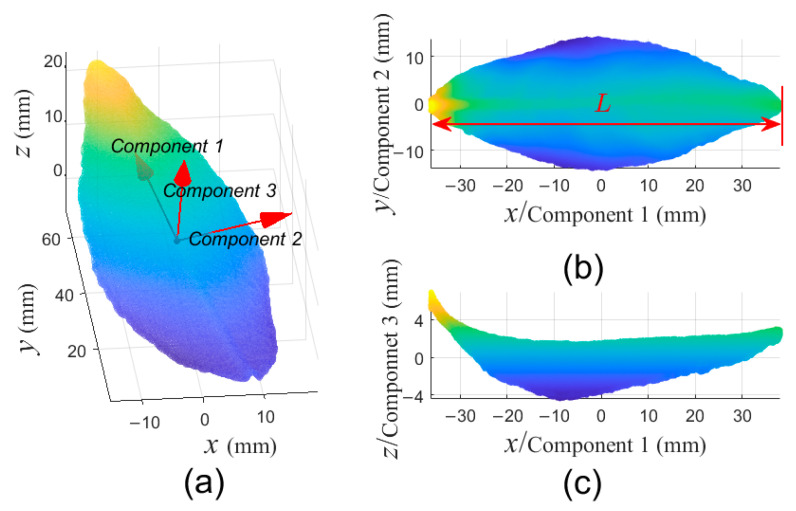
An example of principal component analysis (PCA) for tea leaf point cloud. (**a**) A tea leaf point cloud in cartesian coordinate system, and three principal components represented as red arrows; (**b**) The standard leaf point cloud (SLPC) from view of the third principal component, and the *L* is defined as the length of leaf; (**c**) The SLPC from the view of the second principal component. Component 1, 2, and 3 correspond to the first, second, and third principal components of the point cloud, respectively.

**Figure 4 sensors-21-01304-f004:**
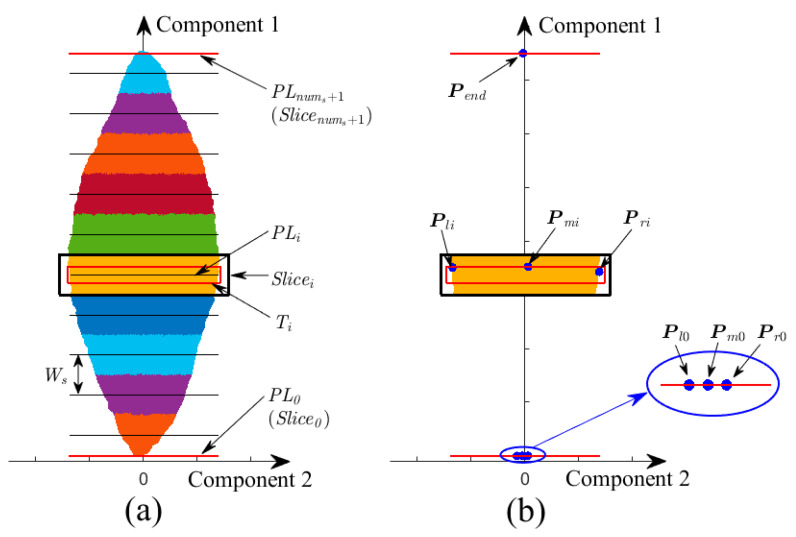
(**a**) An example for slicing point cloud into multi-sections; (**b**) Schematic diagram of extracting feature points belonging to midrib and margin of leaf in different slices. Components 1 and 2 are the first and second principal components of the leaf point cloud, respectively.

**Figure 5 sensors-21-01304-f005:**
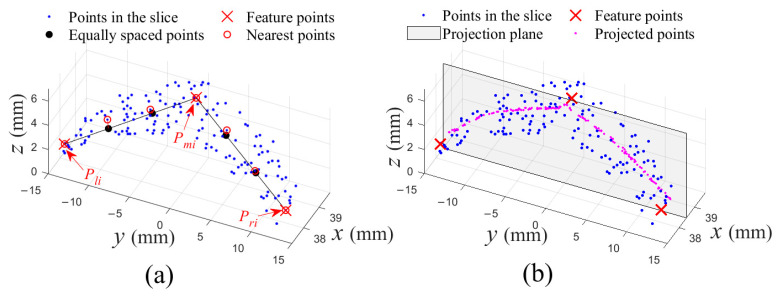
Two generating methods for PSF. (**a**) Seven points are selected as examples in a slice, and $P_{li}$, $P_{mi}$ and $P_{ri}$ are three feature points in the slice; (**b**) All points in a slice are projected vertically on the central plane.

**Figure 6 sensors-21-01304-f006:**
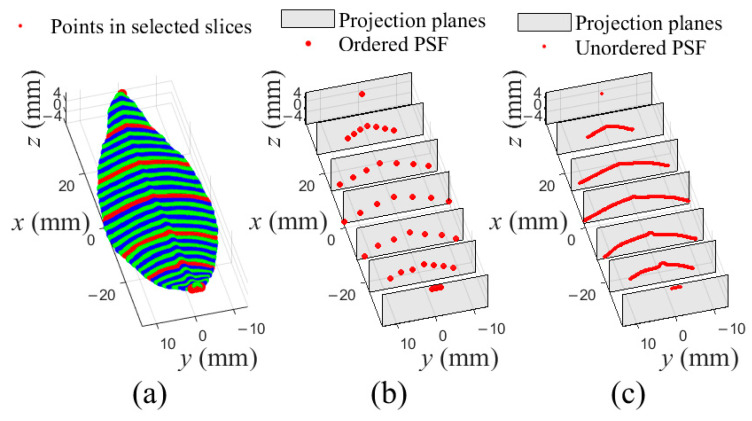
An example of slicing SLPC and generating two kinds of PSF from 7 selected slices. (**a**) Slicing the SLPC into 50 slices, and 7 selected slices marked in red were selected to generate PSFs; (**b**) An ordered PSF composed of 7×7 points from selected slices; (**c**) An unordered PSF generated by projecting points of selected sections onto corresponding center planes.

**Figure 7 sensors-21-01304-f007:**
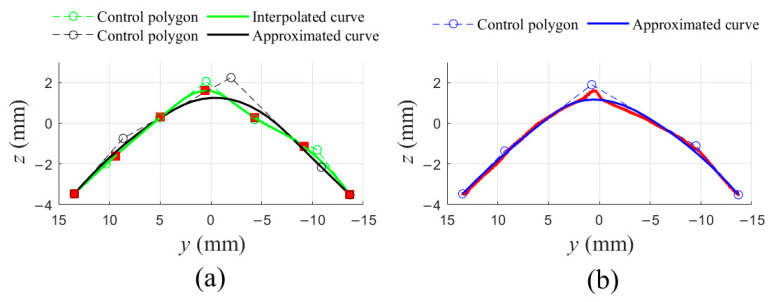
Different fitting curves in a slice. (**a**) 7 points were fitted into an interpolated curve (green curve) and an approximated curve (black curve), respectively; (**b**) All projected points were approximated into one curve.

**Figure 8 sensors-21-01304-f008:**
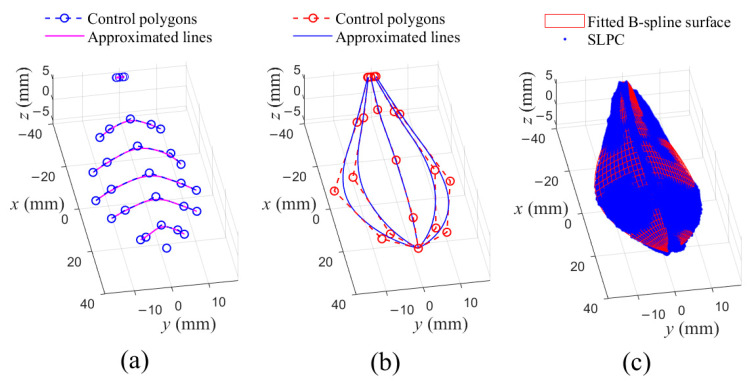
Fitting procedure for the unordered PSF. (**a**) B-spline curve approximations in u-direction; (**b**) B-spline curve approximations in v-direction; (**c**) Fitted B-spline surface for the unordered PSF and the SLPC.

**Figure 9 sensors-21-01304-f009:**
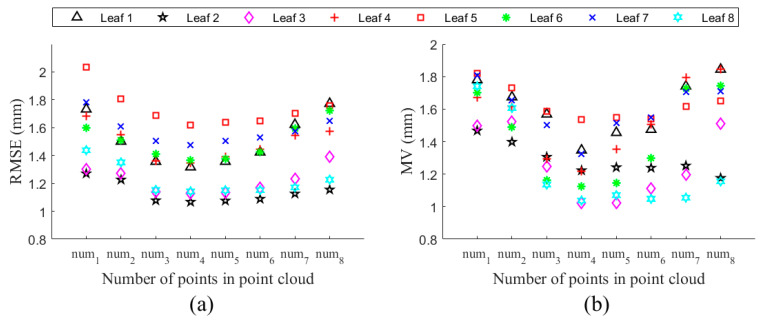
Impacts of the number of points in leaf point cloud on fitting error. (**a**) The RMSE of fitted B-spline surfaces; (**b**) The MV of the fitted B-spline surfaces. ${\mathrm{num}}_{1}$, ${\mathrm{num}}_{2}$, ${\mathrm{num}}_{3}$, ${\mathrm{num}}_{4}$, ${\mathrm{num}}_{5}$, ${\mathrm{num}}_{6}$, ${\mathrm{num}}_{7}$ and ${\mathrm{num}}_{8}$ are equal to 300, 500, 1000, 2000, 4000, 8000, 16,000 and 32,000, respectively.

## Data Availability

Not applicable.

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
