# Peer review of "Parametric Surface Modelling for Tea Leaf Point Cloud Based on Non-Uniform Rational Basis Spline Technique"

_sensors, 2021, doi:10.3390/s21041304_

Round 1

Reviewer 1 Report

General comments: This is a very good manuscript, well written and organized. The introduction and the methodology are all to the point. However authors have used unnecessary sub-sections which I beleive could be merged as one or two main sections. For example, we have sub-section 2.3.2 followed by a number 1. Definition of NURBS Surface. This is supposed to be a scientific journal, not a text-book. Please avoid teaching-style format. All sub-sections that have such a voice should be reworded to match a scientific tone. Other than that, the manuscript can be accepted after minor improvements. 

Title: I would suggest avoiding the use of uncommon abbreviations in the title. The acronyms NURBS is not widely known. Tiel can be reworded as below:

Parametric Surface Modelling for Tea Leaf Point Cloud Based on Non-uniform rational basis spline Technique

Abstract: 

The abstract begins with a valid background and motivational statement about acquisition and segmentation methods of leaf point cloud and the need for parametric surface modelling method. The summary of the method mentioned in the abstract is legit, however authors have failed to highlight the core findings of their approach. What do you mean by "the proposed modeling method could be successfully used"? Please be specific. Is your approach a valid solution under different conditions? If not please highlight the potential and the weak points. 

Keywords: Please remove the acronyms 

Introduction:

  • A literature review is fine and the citing references are all up to date. 
  • The opening statement, background, problem statement, and objective statement are all defined and clearly organized. 
  • Line 31-32: is an excellent justification. Please provide one or two lines stating how leaf morphological features can be utilized in estimating crop yield?

Materials and methods:

2.1. Data collection needs to be expanded. 

Authors need to provide meta-data quality for Fig.2-Fig3...Fig8 (Currently only Fig1 is metadata quality). 

Line 106, Line 121, Line 157 (where you have headings) are creating confusion. Especially after sub-section 3 (line 213) you then have the man section 3 (Results). 

  • Too much use of abbreviations in the headings and titles. 
  • Figure 8: The x-axis says "number of points", but you have labeled as num1, num2...Please clarify. Is your x-axis representing the label of the point? When you say the number of points, then it means points are increasing...and it means your variable is discrete, not continuous. Then why do you have your points connected with each other with a solid line? There is no value between num1 and num2! You can only use a solid line if your x-axis is a continuous variable.

The conclusion needs improvement.

Reviewer 2 Report

This paper presents a preliminary study of the parametric surface modeling method for accurately fitting tea leaf point cloud into NURBS surface. The point cloud data captured using a handheld 3D scanner. The methodology is described well, and the selected experiments are appropriate for validating the proposed method.

The manuscript is interesting, and the results support the claims of the authors. The paper is well organized and presented. There are, however, some minor aspects that the authors should check:

  • The abstract need more discussion on obtained results.
  • The introduction offers enough information. However, it is good to include a figure in the introduction section showing the overview/ summary of the proposed work. It will grab the attention of the researcher in this field.
  • Suggested to include related work.
    • “Chambelland, Jean-Christophe, et al. "A double-digitising method for building 3D virtual trees with non-planar leaves: application to the morphology and light-capture properties of young beech trees (Fagus sylvatica)." Functional Plant Biology 35.10 (2008): 1059-1069.”
    • Beardsley and G. Chaurasia, "Editable Parametric Dense Foliage from 3D Capture," 2017 IEEE International Conference on Computer Vision (ICCV), Venice, 2017, pp. 5315-5324, doi: 10.1109/ICCV.2017.567.
  • Suggested to maintain consistency in the figure’s sub numbering fonts (Figure 2 (a) (b) (c)).
  • Cross-check equation 7.
  • Regarding the conclusion, it is quite brief and can be extended in a more detailed assessment of the gained results.
